# Implementation of Underwater Electric Field Communication Based on Direct Sequence Spread Spectrum (DSSS) and Binary Phase Shift Keying (BPSK) Modulation

**DOI:** 10.3390/biomimetics9020103

**Published:** 2024-02-09

**Authors:** Yuzhong Zhang, Zhenyi Zhao, Xinglong Feng, Tianyi Zhao, Qiao Hu

**Affiliations:** 1School of Mechanical Engineering, Xi’an Jiaotong University, Xi’an 710049, China; zyz_backtoblack@stu.xjtu.edu.cn (Y.Z.); zzyxjtu@stu.xjtu.edu.cn (Z.Z.); 2Shaanxi Key Laboratory of Intelligent Robots, Xi’an Jiaotong University, Xi’an 710049, China; fengxinglong@stu.xjtu.edu.cn (X.F.); zty_xjtu@stu.xjtu.edu.cn (T.Z.)

**Keywords:** weak electric fish, direct sequence spread spectrum, binary phase shift keying modulation, electric field communication

## Abstract

Stable communication technologies in complex waters are a prerequisite for underwater operations. Underwater acoustic communication is susceptible to multipath interference, while underwater optical communication is susceptible to environmental impact. The underwater electric field communication established based on the weak electric fish perception mechanism is not susceptible to environmental interference, and the communication is stable. It is a new type of underwater communication technology. To address issues like short communication distances and high bit error rates in existing underwater electric field communication systems, this study focuses on underwater electric field communication systems based on direct sequence spread spectrum (DSSS) and binary phase shift keying (BPSK) modulation techniques. To verify the feasibility of the established spread spectrum electric field communication system, static communication experiments were carried out in a swimming pool using the DSSS-based system. The experimental results show that in fresh water with a conductivity of 739 μS/cm, the system can achieve underwater current electric field communication within a 11.2 m range with 10^−6^ bit errors. This paper validates the feasibility of DSSS BPSK in short-range underwater communication, and compact communication devices are expected to be deployed on underwater robots for underwater operations.

## 1. Introduction

The complex underwater environment makes underwater operations very challenging. Additionally, the most commonly used electromagnetic waves undergo severe attenuation underwater and are almost unusable. Deploying an underwater wireless communication system in water presents more challenges compared to a ground-based wireless communication system.

Underwater acoustic communication (UWAC) is one of the commonly used underwater communication methods. At present, UWAC can support transmission rates of up to hundreds of bits per second over ranges of up to hundreds of kilometers [1]. UWAC utilizes mechanical waves as the carrier supplemented with transducers and other devices for communication. According to the signal propagation characteristics in the underwater acoustic channel, when two devices equipped with UWAC are located in different sound velocity layers, it is easy to cause blind zones [2,3]. The transmission speed in the underwater acoustic channel is only about 1500 m/s, resulting in large propagation delays. In small and confined underwater areas, UWAC is severely affected by multipath effects and shallow water noise, making it unsuitable for short-range communications. At the same time, UWAC requires mutual conversion between electric and acoustic energy, so most devices are large-sized, making them difficult to supply for small autonomous underwater vehicles or divers. Therefore, in applications requiring high communication rates, low latency, short-distance communications, and device miniaturization, UWAC is not very suitable.

Underwater optical communication (UWOC) is another common underwater wireless communication method. According to research, most lasers would be absorbed by seawater and unable to penetrate the ocean, but blue–green lasers (wavelengths approximately 470–570 nm) have the lowest attenuation when propagating through seawater [4,5]. Compared to UWAC, UWOC has advantages like a high transmission bandwidth, fast communication data rate, and low link latency. The development of UWOC has been relatively rapid, with huge potential. However, there are some problems that need to be solved before it can be widely applied [6]. First, UWOC is susceptible to turbidity, turbulence, bubbles, suspended solids, and other hydrological conditions that can scatter light in water, requiring high environmental demands on the water body. Second, precise alignment is required between the optical transmitter and receiver, with most systems currently only able to communicate within line-of-sight ranges. Due to these reasons, the application scenarios of UWOC are relatively narrow at present. Therefore, how to accurately detect targets and ensure smooth information interaction between underwater swarm vehicles in complex underwater working environments has become a technical challenge for underwater vehicle applications.

Research has found that weakly electric fish living in Africa generate electrical signals through a specialized discharge organ and perceive the surrounding environment through receptors distributed on their skin [7]. These fish have the ability to sense electric fields, enabling them to navigate underwater, recognize targets, and engage in intraspecific communication [8]. Inspired by this phenomenon, Joe et al. [9] built an experimental circuit for quasi-static electric field digital signal transmission, achieving high-bitrate digital information transfer over short distances. They further established a distance–frequency-dependent path loss model applicable for frequencies up to 1.2 MHz and distances up to 5 m [10]. Friedman et al. [11,12] showed that electromagnetic radiation propagating upward, then laterally near the surface before descending directly to the receiver is often optimal. Rauf et al. [13,14] experimentally and via simulations demonstrated that low bit error rates are only observed when node immersion depths exceed dipole lengths. Makinwa et al. [15] analyzed underwater communication using quasi-static electric fields, validating the quasi-static model through measurements. Zhao et al. [16] wirelessly transmitted text and images at 16 bits/s underwater using Maxwell displacement currents from triboelectric nanogenerators.

Herein, we established an underwater electric field communication system based on biomimetic principles and explored ways to improve its communication reliability. The remainder of this paper is organized as follows. Section 2 describes the theory of underwater electric field communication and performs underwater electric field modeling and simulation using CST 2021 software to determine the parameters for underwater electric field communication. Guided by the theoretical parameters from the previous section, Section 3 presents the design of a DSSS BPSK underwater electric field communication system; one significant advantage of DSSS technology is its strong resistance to interference. Section 4 provides comparative experiments and discusses the experimental results. Finally, Section 5 summarizes the conclusions. The results indicate that the constructed electric field communication system qualitatively validated the simplified discharge organ model and achieved error-free communication at a rate of 11.2 m/1.2 kHz under low power consumption of 5 W. Low-power and compact systems are expected to be mounted on small underwater robots for operations.

## 2. Theory of Underwater Electric Field Communication

### 2.1. Analysis of Electric Field Sensing Mechanism of Weak Electric Fish

Different fish use various communication and perception methods, including vision, hearing, chemistry, and the lateral line sensory system. Some fish can generate weak electrical signals using their own organs and perceive these signals, allowing them to sense their environment and engage in information exchange. The distribution of these organs is indicated by the red area in Figure 1a. These weakly electric fish create an electric field by repeatedly emitting electric organ discharges (EOD) through their electric organs (EO), which polarize surrounding objects. Polarized objects alter the distribution of the electric field, and weakly electric fish detect changes in the electric field to perceive their surroundings [17,18].

The EOD waveforms generated by weakly electric fish consist of pulse signals lasting from 100 μs to 20 ms, as illustrated in Figure 1b, and their propagation underwater is nearly instantaneous with minimal delay. The electric organ of weakly electric fish is typically located in the tail, while electric signal receptors are found in the head and trunk, as depicted in Figure 1c. These electric receptors guide the current into sensory epithelial cells, and the electric signal subsequently stimulates the central nervous system to produce appropriate responses.

### 2.2. Electric Field Communication Model

As the electric field communication mechanism in weakly electric fish is complex, the fish emitting signals generate an electric field through a large array of cells on their bodies, and the receiving fish sense this electric field. They decode the electric signals using thousands of electric receptors distributed on their skin. Therefore, in order to develop a communication system suitable for underwater robots, the biological communication system of weakly electric fish is simplified into a model with a pair of transmitting electrodes and a pair of receiving electrodes, as shown in Figure 1d. The electric field generated by weakly electric fish is essentially a low-frequency alternating electromagnetic wave. During the process of underwater electric field communication, the generation of this electric field can be simulated by emitting continuously varying currents through a pair of electric dipoles.

Underwater channels are also good conductors, and the changing potential on the electric dipoles results in varying currents in the water, consequently generating alternating electromagnetic fields. Therefore, the influence of both conduction current and displacement current needs to be considered.

Specifically, conduction current is formed due to the movement of charges between atoms and is time-invariant. It follows Ohm’s law, and its expression is given by:(1)$$J_{c}=\sigma E$$
where σ—the conductivity S/m; E—the electric field strength V/m; $J_{c}$—the conduction current density ${A/m}^{2}$. The displacement current is a time-varying current generated by the motion of bound charges within atoms induced by an external field, and its density is expressed as:(2)$$J_{d}=\varepsilon \frac{\partial E}{\partial t}$$
where ε—the permittivity. Usually, at 20 °C, the permittivity of water is 80 times that of a vacuum ($\varepsilon_{0}=8.85\times 10^{-12}\;F\cdot m^{-1}$).
(3)$$J_{d}=-\omega \varepsilon E_{0}\sin\omega t$$

In the equation, $E_{0}$—amplitude of the alternating electric field; ω—angular frequency of the alternating electromagnetic field. Likewise, the conduction current density $J_{c}$ also has a similar form.
(4)$$J_{c}=\sigma E=\sigma E_{0}\cos\omega t$$

As mentioned in the previous section, the electrical signal used by weakly electric fish for electric field communication is a quasi-static electric field. To meet the quasi-static field conditions, the field used for transmission should be primarily composed of a time-invariant conduction current. Therefore, the conduction current density and displacement current density for this alternating electromagnetic field should satisfy the following:(5)$$\frac{\left|J_{d}\right|}{\left|J_{c}\right|}=\frac{\varepsilon }{\sigma }\omega <<1$$

Typically, in seawater, $\sigma =3\sim 5\;Sm^{-1}$, while in freshwater, $\sigma =0.01\sim 0.1\;Sm^{-1}$. In practical engineering applications, the prevailing condition for conduction current to be the primary component of the total current is $\frac{\left|J_{d}\right|}{\left|J_{c}\right|}<0.1$. As per Equation (5), in a seawater environment, when the frequency of the alternating electric field is below 230 MHz, the time-varying displacement current can be neglected. However, in a freshwater environment, the frequency of the alternating electric field needs to be lowered to a few hundred kHz. When constructing an electric field communication model and designing a system, it is essential to consider the frequency of the alternating electric field.

In the simplified electric dipole model, the field strength E at the receiver r is:(6)$$E=E_{r}+E_{\theta }+E_{\varphi }$$
(7)$$E_{r}=\frac{I_{0}d_{1}e^{-\gamma r}}{4\pi \sigma r^{3}\left(\sigma +j\omega \varepsilon \right)}\left(1+\gamma r\right)2\cos\theta_{1}\cos\theta_{2}$$
(8)$$E_{\theta }=\frac{I_{0}d_{1}e^{-\gamma r}}{4\pi \sigma r^{3}\left(\sigma +j\omega \varepsilon \right)}\left(1+\gamma r+\gamma^{2}r^{2}\right)\sin\theta_{1}\sin\theta_{2}$$
(9)$$E_{\varphi }=0$$
where σ is the electrical conductivity of the medium, ε is the permittivity of the medium, μ is the magnetic permeability, and $\gamma =\sqrt{j\omega \mu \left(\sigma +j\omega \varepsilon \right)}$.

When the receiver is on the axis of symmetry of the electric dipole ($\theta =\frac{\pi }{2}$, φ=0), the electric field strength is:(10)$$E=E_{\theta }=\frac{I_{0}d_{1}}{4\pi \sigma }\frac{1}{r^{3}}\left(1+\gamma r+\gamma^{2}r^{2}\right)e^{-\gamma r}$$

Therefore, when the receiving electrodes are separated by a distance *d*_2_, the potential difference received is:(11)$$U=\int_{0}^{d_{2}}Eds=\frac{I_{0}d_{1}d_{2}}{4\pi \sigma }\frac{1}{r^{3}}\left(1+\gamma r+\gamma^{2}r^{2}\right)e^{-\gamma r}$$

The frequency of the electric field changes is inversely correlated with the intensity of the electric field strength at the receiving end. In the communication system, it is necessary to consider both the communication distance and communication rate simultaneously to determine an appropriate carrier parameter.

### 2.3. Electric Dipole Field Strength

Simulating underwater electric field application scenarios and validating the effect of frequency on electric field propagation helps determine the optimal communication frequency. Theoretical modeling, solving, and simulation analysis were performed using CST 2021 software. Based on the biomimetic principles, underwater electric field communication is a low-frequency electromagnetic field. Therefore, we utilized the CST EMSTUDIO software with a low-frequency time-domain solver for simulation. The electrode material was set to PEC (Perfectly Electric Conducting) with infinite conductivity. The dipole was symmetrically distributed on both sides of the origin on the *x*-axis, with the background environment set to freshwater. Table 1 displays the specific parameter settings.

In order to analyze the electromagnetic field propagation characteristics of the dipole in the underwater electric field communication model, we simplified the analysis of the electric field strength along the perpendicular axis of the dipole. Due to the symmetry of the dipole’s electric field, the analysis and calculations in this paper focused on one side of the model. The meshing of the model was conducted using tetrahedral elements, with mesh size tailored to the field strength. Densely packed meshes were used in regions of strong electric field, while sparse meshes were applied in areas with weaker field strength. This mesh strategy accelerated the solver’s solution process and ensured a smoother data curve. Additionally, the grids near the perpendicular axis of the dipole were refined to guarantee accurate output at the axis. The simulation model employed an adaptive meshing approach. Through the software’s adaptive mesh solver, the grid size was refined to obtain more precise simulation data.

The distribution of the electric field strength was generated by the electric dipoles in a freshwater environment. The electric field strength distribution exhibits symmetry with respect to the *xoz* plane (y = 0) and the *xoy* plane (z = 0). The attenuation of the electric field strength with increasing distance is observed when different frequency signals are applied to the electric dipoles. The signal frequencies used are 30 KHz, 50 KHz, 55 KHz, and 60 KHz.

Based on the analysis of the results in Figure 2, it is observed that the electric field strength at the same location undergoes a sharp attenuation when the frequency of the alternating electric field is above 50 kHz. The alternating electric field at 50 kHz experiences relatively small attenuation compared to the 30 kHz alternating electric field, and the signal has a higher bandwidth. Considering the rapid attenuation of the field strength with frequency and the communication system’s communication rate requirements, a carrier frequency of 50 KHz is selected.

## 3. DSSS in Electric Field Communication

Spread spectrum technology involves the use of a specific spreading function. This function expands the spectrum of a signal carrying information to be transmitted. The result is the transformation of the signal into a wideband signal. The signal is then modulated onto a carrier and transmitted through the channel. At the receiving end, a correlation-based process is used to compress the spread spectrum, thus recovering the original baseband signal’s bandwidth and enabling information transmission [19]. In the spread spectrum system, the signal transmitted through the channel has a wider bandwidth than the original signal. Spreading is achieved by using pseudorandom codes that are independent of the information, expanding the signal’s spectrum significantly beyond that of the original signal. Due to the spread spectrum communication technology expanding the signal spectrum, it possesses strong resistance to narrowband interference and noise. DSSS is a type of spread spectrum technology, and compared to other spread spectrum techniques, it exhibits stronger resistance to interference.

### 3.1. The Modulation of DSSS BPSK Signals

In a digital communication system, the source output contains the information to be transmitted, known as the digital baseband signal. To transmit the signal through an underwater channel, the digital signal must be modulated onto a carrier wave, shifting the signal’s spectrum to a higher frequency. Modulation is the process of using the baseband signal to control certain parameters of the carrier wave. Digital modulation can be divided into Amplitude Shift Keying (ASK), Frequency Shift Keying (FSK), and Phase Shift Keying (PSK). For a digital communication system, the bit error rate (BER) is an important metric to assess its reliability and effectiveness. The BER formulas for binary digital modulation systems are shown in Table 2 [20].

where r represents the SNR, and erfc(x) is the complementary error function, expressed as follows:(12)$$erfc(x)=\frac{2}{\sqrt{\pi }}\int_{x}^{\infty }e^{-\eta^{2}}d\eta$$

Comparatively, under the same SNR, the BER is the lowest when using BPSK modulation, followed by 2FSK, and 2ASK has the highest BER. For the same modulation scheme, coherent demodulation results in a lower BER compared to non-coherent demodulation. Therefore, BPSK modulation offers better interference resistance than 2ASK and 2FSK. This paper employs BPSK modulation with coherent demodulation.

BPSK is a modulation process that utilizes changes in the carrier wave’s phase to convey digital signals. In BPSK modulation, the carrier wave’s frequency and amplitude remain constant. When transmitting bit “1”, the carrier wave phase is set to $\varphi_{1}$, and for bit “0”, the carrier wave phase is $\varphi_{2}$. This can be represented as:(13)$$e_{2psk}=\left\{\begin{array}{cc} A\cos(\omega t+\varphi_{1}) & \mathrm{When}\;“1”\;\mathrm{is}\;\mathrm{sent}\;\mathrm{with}\;a\;\mathrm{probability}\;\mathrm{of}\;p\\ A\cos(\omega t+\varphi_{2}) & \mathrm{When}\;“0”\;\mathrm{is}\;\mathrm{sent}\;\mathrm{with}\;a\;\mathrm{probability}\;\mathrm{of}\;(1-p) \end{array}\right.$$

Its waveform is depicted in the following Figure 3:

The receiver demodulates the received BPSK signal to recover the baseband signal using coherent demodulation, employing the Costas loop [21,22]. The principle of DSSS modulation is to modulate the original signal with a random sequence before carrier modulation. In practical use, considering the difficulty of replicating a truly random sequence, m-sequences are often used as pseudorandom code sequences to spread the signal. Based on the simulation in Section 2, the parameters used are shown in Table 3.

After determining the primitive polynomial, a linear shift register can be constructed based on this primitive polynomial. Using a fifth-degree primitive polynomial generates a periodic m-sequence with a period of 31. The modulation signal in the DSSS can be divided into three main processes: generating a pseudorandom code sequence, spreading the original signal through the pseudorandom code sequence, and modulating the carrier. Meanwhile, pulse shaping is applied to the spread signal to filter out signals and noise outside the main lobe, improving transmission efficiency. The phase transition of a PSK signal is relatively gradual at the symbol transition instants. The waveform and spectrum of a directly spread modulated signal are shown in Figure 4.

### 3.2. The Demodulation of DSSS BPSK Signals

At the receiving end, the received spread spectrum signal undergoes high-frequency amplification and mixing. It is correlated with a pseudorandom code sequence synchronized with the transmitter to despread the intermediate frequency spread modulation signal, restoring the spectrum of the spread modulation signal to that of the intermediate frequency modulation signal. Then, demodulation is performed to recover the transmitted baseband signal, completing the information transmission.

A spread spectrum BPSK system builds upon baseline BPSK modulation and demodulation by incorporating spreading and despreading functions. The demodulation circuit still requires a Costas loop for carrier recovery [23]. The Costas loop, employed in the extraction of coherent carriers, also accomplishes the demodulation of baseband signals. Upon extracting the demodulated baseband signal, synchronization of the symbol clock for information bits can be associated with the pseudocode symbol clock. The synchronization of one naturally leads to the synchronization of the other. A pseudorandom code despreading module is added to the Costas loop so that the receiver can achieve bit synchronization, Ref. [24] pseudorandom code synchronization, and carrier synchronization with the transmitter, as shown in Figure 5.

Pseudocode synchronization consists of two stages: capture and tracking. In the acquisition stage, the local signal is correlated with the received signal to obtain correlation values. A threshold detection is used to determine whether the useful signal has been acquired. In the specific implementation, when the received pseudorandom code sequence deviates in frequency from the local pseudorandom code sequence, the two sequences slide relative to each other in phase. Upon the phases of the two sequences coinciding at a certain moment, i.e., achieving phase alignment, the correlation value exhibits its maximum output. The sliding is halted at this point, marking the completion of the capture operation. Once the useful signal is acquired, and the phase difference between the pseudocodes at the transmitter and receiver is within one pseudocode chip, the system enters the tracking stage.

The capture and tracking stages employ distinct loops, with the two stages following a sequential order on the locking loop. This theoretical framework allows for the reuse of integrators, local pseudocode generators, and multipliers between the two loops, thereby minimizing the logical resource consumption in processing units. Paper [25] proposes a structure that reuses certain components between the capture and tracking loops. The primary principle involves using the squared sum of the leading and lagging branch accumulators from the delay-locked loop as a criterion for determining the presence of a useful signal. While the paper illustrates a flat sum of the peak outputs of the two branch accumulators, in practical terms and under the system’s parameters, the curve of the sum of peaks exhibits a concave shape. This variation in the sum of peaks is unfavorable for determining whether the tracking loop is locked. Therefore, as shown in Figure 6, this paper introduces an additional correlator on the intermediate branch above the leading and lagging branches, serving as the decision basis for the capture state.

### 3.3. Design of the Electric Field Communication System

In the pursuit of achieving a compactly integrated underwater electric field communication system, we present the design and implementation of the communication system depicted in Figure 7. All relevant calculations for digital signal processing are executed using an FPGA.

The hardware platform of the system is depicted in Figure 8, comprising a PC (including Raspberry Pi), an FPGA, and an analog circuit. In this configuration, the Raspberry Pi serves as the terminal for the communication system, allowing users to send source information to the FPGA via a serial port. The FPGA functions as the digital signal processing and acceleration platform for the communication system, facilitating data transfer with the analog circuits through A/D and D/A.

The transmitting end is composed of a transmission system and transmitting electrodes. The PC host sends information to the transmitting FPGA host through UART. The structure of the transmitter system is illustrated in Figure 9. Within the FPGA, channel encoding is employed as the initial step to combat white noise. A (15,7) Reed–Solomon channel code is utilized, incorporating eight parity bits into the information code to correct up to four symbol errors [26].

The generation of the pseudocode sequence is a crucial step in direct sequence modulation systems. The pseudocode generator can be implemented based on a fifth primitive polynomial, constructing a linear feedback shift register with a period of 31. The initial values and tap positions of the shift register are set to fixed values. Under clock drive, a complete cycle of the pseudocode is sequentially read out. The original signal is then spread by the pseudocode signal, with one data symbol period aligning with one pseudocode period, ensuring that a complete pseudocode period precisely encompasses one original data symbol.

The receiving end primarily consists of receiving electrodes, a receiving processor, and a PC. The electric field information generated by the transmitting electrodes is sensed using a pair of titanium electrodes. The structure of the receiver system is shown in Figure 10. At the forefront of the receiving system is a first-stage low-noise Automatic Gain Control (AGC) [27] amplifier circuit. The amplified signal is then converted into a digital signal through an Analog-to-Digital Converter, and subsequently transmitted in parallel to the FPGA for digital signal processing.

The demodulation system of the direct sequence system is essentially a BPSK demodulation system. The demodulation circuit continues to use the Costas loop; however, it requires the addition of a pseudocode synchronization step within the Costas loop. The pseudocode synchronization module, as detailed in Section 3.2, is designed specifically for this purpose and can be directly invoked as a file in the top-level hierarchy. The despread signal enters the modem for demodulation and decoding, thereby recovering the transmitted signal.

The decision outcome of the signal is decoded to obtain the original baseband data, which is then transmitted via UART to the PC for printing and error rate counting.

## 4. Experimental Verification

To verify the performance of this communication system based on DSSS, static underwater experiments were carried out. The experimental equipment included two communication systems, transmitting and receiving electrodes. Since the experimental environment could not simulate an infinite unbounded water area, a 50 m × 15 m × 2 m swimming pool was chosen based on the actual situation. The measured electrical conductivity of the pool water was 739 μS/cm.

As shown in Figure 11a, we experimented with electrode patches of different sizes, as shown in Figure 11b. By examining the potential difference of the receiving electrodes through an oscilloscope, we confirmed that using 100 mm × 100 mm × 0.2 mm transmitting electrodes and 80 mm × 80 mm × 0.2 mm receiving electrodes in the experiment results in the maximum received signal-to-noise ratio (SNR). Considering that this system will be installed on underwater robots in the future, the spacing between the transmitting electrodes is set to 60 cm, and the spacing between the receiving electrodes is also set to 60 cm to accommodate the size of underwater robots.

The signal received by the receiving electrode undergoes digital signal processing in the FPGA, and the relevant signals are captured using a logic analyzer. Figure 12 illustrates the demodulation, despreading, and decision-making processes performed after the receiver receives the signal.

The experimental scene is shown in Figure 11d. Firstly, we verified the impact of electrode angles on the electric field communication system and validated the dipole model. In this specific experiment, while keeping the receiving electrode stationary, we altered the angle between the transmitting electrode and the receiving electrode to evaluate the effect of different angles on communication performance. Experiments were conducted by rotating the transmitting electrode counterclockwise in 5° increments, and the communication error rate was measured. The experimental results are presented in Figure 13. When the angle was between 0° and 25° and between 145° and 180°, the communication error rate was 100%. At 30°, it was 7.88%, at 35° it was 18.43%, at 40° it was 39.26%, at 45° it was 60.05%, and at 50° it was 97.67%. As the angle between the transmitting and receiving electrodes approached parallel alignment (90°), the error rate gradually decreased. Beyond 90°, as the angle approached perpendicular alignment, the error rate gradually increased. At 125°, it was 96.25%, at 130° it was 61.72%, at 135° it was 21.58%, and at 140° it was 7.15%. For all other angles, the error rate was 0%. The experimental results indicate that the highest communication efficiency is achieved when the transmitting and receiving electrodes are parallel (0°, 180°), and the error rate reaches 100% when they are perpendicular (55°~120°). According to the theory of electric field communication, the induced electromotive force is the potential difference generated at the receiving electrode by the electric field produced by the transmitting electrode. When both pairs of electrodes are parallel, the potential difference generated is at its maximum, and when both pairs of electrodes are perpendicular, the potential difference is at its minimum. This result is in good agreement with the simplified dipole model and serves to establish the optimal electrode layout for subsequent static experiments.

Static single-node ranging experiments were carried out to evaluate the underwater communication effects and maximum communication range of this underwater current electric field communication system. As shown in Figure 14, the distance between the transmitting and receiving electrodes was changed. By adjusting the amplification multiples of the power amplifier circuit and front-end automatic gain amplifier circuit, the signal voltage was adjusted to an appropriate level. The BER was calculated based on the number of correct and incorrect information bits received through the PC serial port. When the communication distance was less than 11.2 m, the communication bit error rate was 0.0001%; when the distance was greater than 11.2 m, the bit error rate increased rapidly as the distance increased; when the communication distance was 11.6 m, the communication BER was about 50%; when the communication distance was 12 m, the BER was 99%; and when the communication distance exceeded 12.5 m, the communication bit error rate was 100%. This underwater current electric field communication system can achieve underwater communication within an 11.2 m range with a bit error of 0.0001%. In comparison, under the same transmission power, the communication distance was 2.4 m for the 2ASK modulation and 3.8 m for the FSK modulation. In contrast, this system has a longer communication range at the same bit error rate [28].

## 5. Conclusions

Based on the conductive characteristics of electromagnetic waves in water, we established a mathematical model of a static electric field using a dipole pair, and generalized it to electric field-based communication systems. A DSSS BPSK communication system was designed and verified the feasibility of this approach.

The experiment on the underwater electric field communication angle validated the accuracy of the simplified dipole model. Maximum potential difference occurred when both electrode pairs were parallel, and minimum when perpendicular. The underwater static communication experiments demonstrated that the designed DSSS BPSK wireless communication system could transmit information error-free up to 11.2 m. The DSSS BPSK system spreads the spectrum of interference and noise, thereby reducing interference power within the signal band and enhancing the demodulator’s input signal-to-noise ratio. Compared to previous work, this system achieved a 40% increase in communication range and a 28% decrease in power consumption [28].

The DSSS technique facilitates multi-access communication by allocating unique spreading codes. It leverages the autocorrelation properties of these codes to accurately receive signals from specific users. This approach holds significant potential for application in underwater robot swarm operations.

## Figures and Tables

**Figure 1 biomimetics-09-00103-f001:**
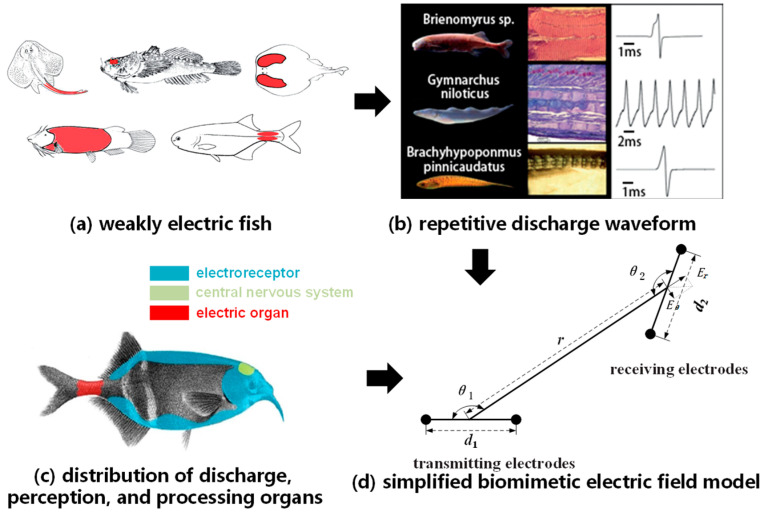
The establishment of a biomimetic electrode model for weakly electric fish discharge and perception.

**Figure 2 biomimetics-09-00103-f002:**
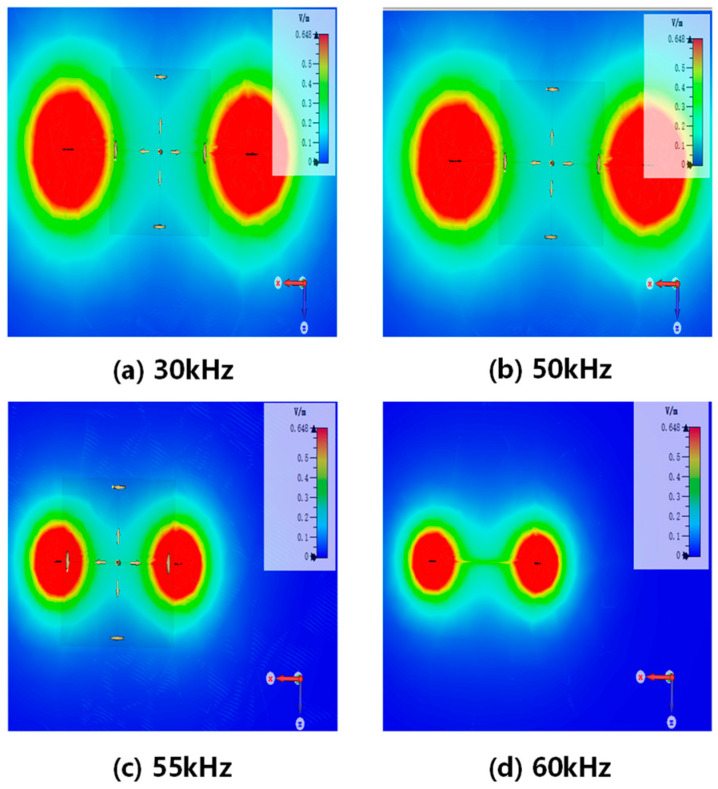
Distribution of electric field strength in the *xoz* plane (*y* = 0) at different frequencies.

**Figure 3 biomimetics-09-00103-f003:**
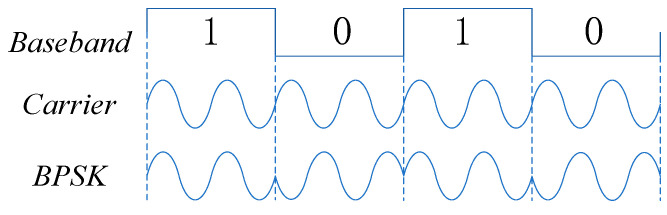
BPSK modulation waveform.

**Figure 4 biomimetics-09-00103-f004:**
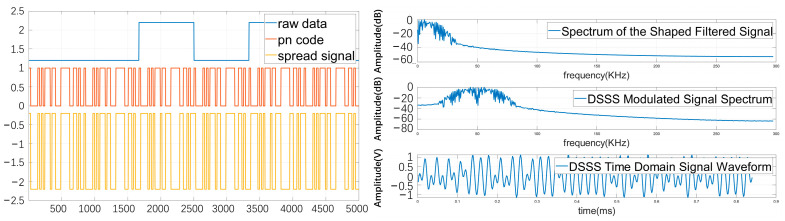
Waveform and spectrum of a DSSS modulated signal.

**Figure 5 biomimetics-09-00103-f005:**
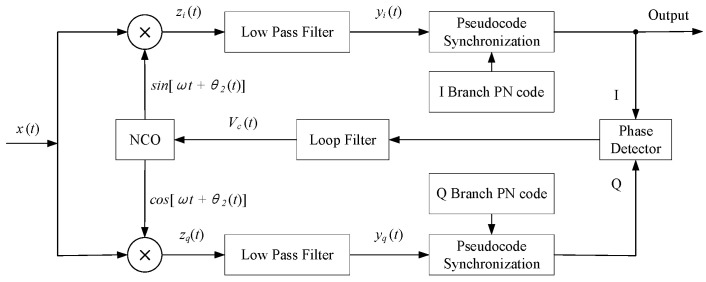
Flowchart of spread spectrum BPSK demodulation.

**Figure 6 biomimetics-09-00103-f006:**
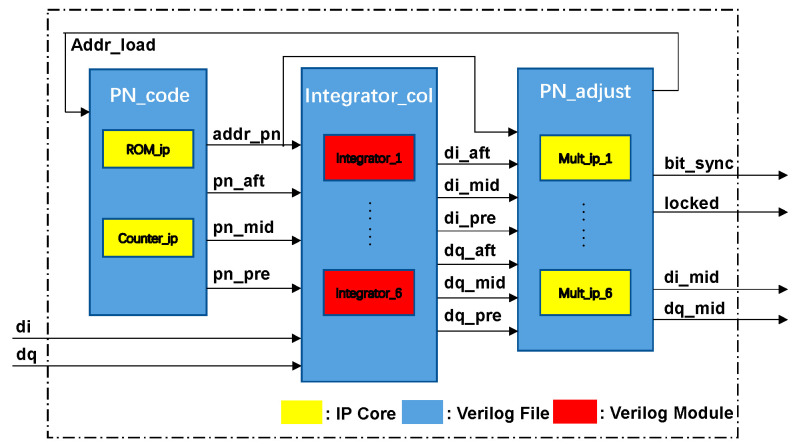
Implementation of pseudocode synchronization in FPGA.

**Figure 7 biomimetics-09-00103-f007:**
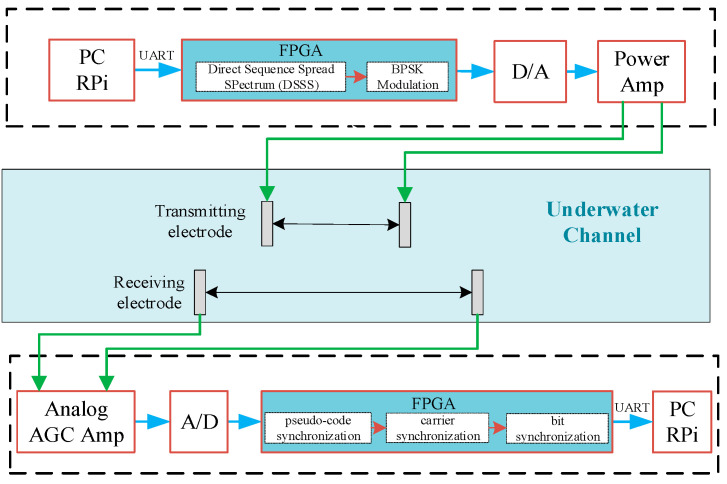
DSSS BPSK communication system.

**Figure 8 biomimetics-09-00103-f008:**
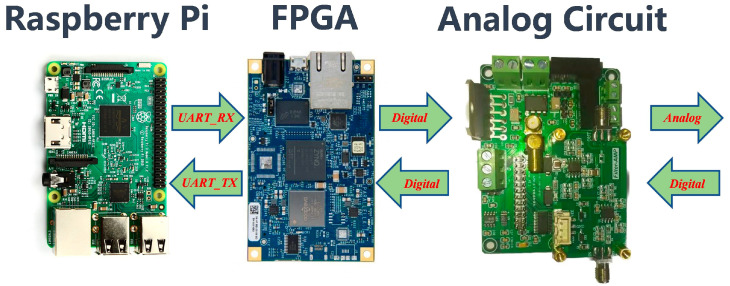
The hardware platform.

**Figure 9 biomimetics-09-00103-f009:**
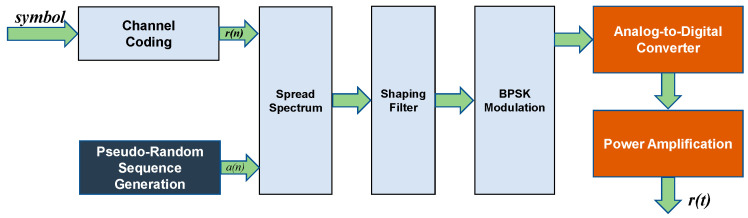
BPSK DSSS transmitter structure.

**Figure 10 biomimetics-09-00103-f010:**
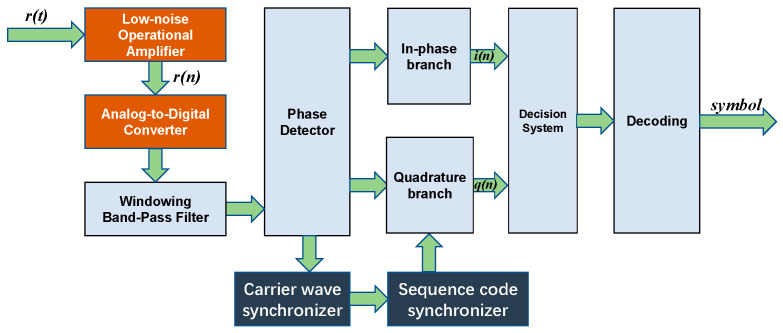
BPSK DSSS receiver structure.

**Figure 11 biomimetics-09-00103-f011:**
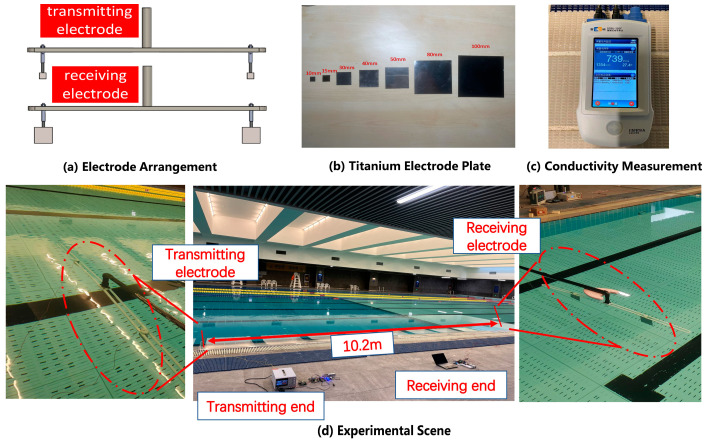
Experimental equipment and experimental setting.

**Figure 12 biomimetics-09-00103-f012:**
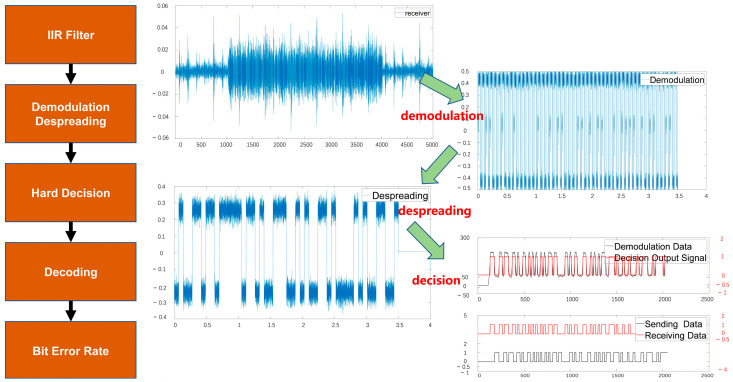
Receiver signal visualization.

**Figure 13 biomimetics-09-00103-f013:**
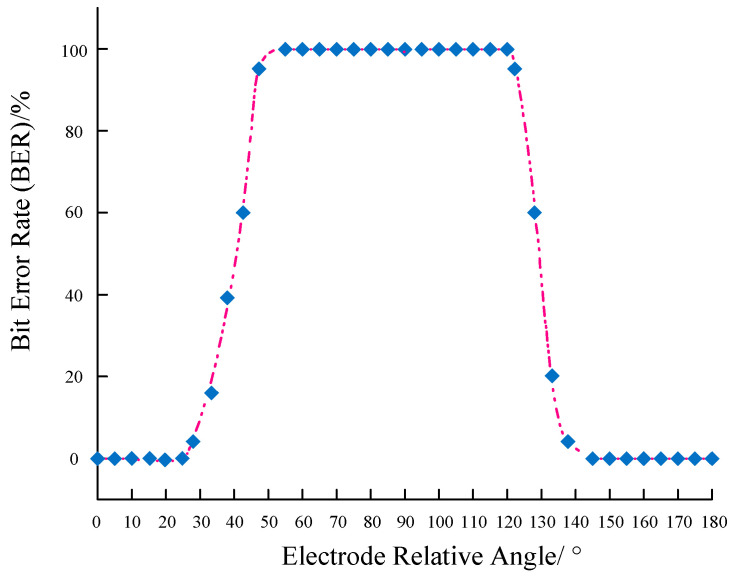
Underwater electric field communication angle experiment.

**Figure 14 biomimetics-09-00103-f014:**
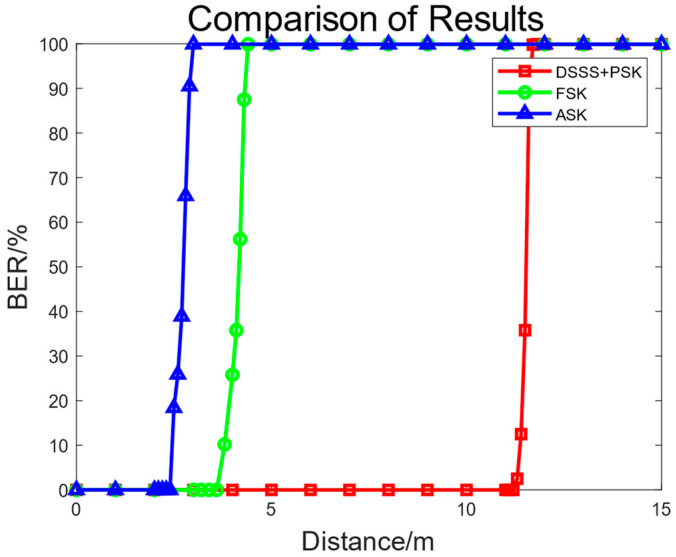
Comparison of communication distances and BER for different modulation schemes.

**Table 1 biomimetics-09-00103-t001:** Freshwater medium and electrode model parameters.

Parameter	Value
Model dimensions	800 m × 800 m × 1000 m
Electrical conductivity	σ=487uS/cm
Relative permittivity	$\varepsilon_{r}=74$
Relative permeability	$\mu_{r}=0.9999$
Electric dipole size	15 mm × 15 mm × 0.2 mm
Distance between electric dipoles	1 m
Current between electric dipoles	0.5 A

**Table 2 biomimetics-09-00103-t002:** BER formulas for binary digital modulation systems.

Modulation	The Coherent Demodulation BER (Pe)	The Non-Coherent Demodulation BER (Pe)
2ASK	$\frac{1}{2}erfc(\sqrt{\frac{r}{4}})$	$\frac{1}{2}e^{\frac{-r}{4}}$
2FSK	$\frac{1}{2}erfc(\sqrt{\frac{r}{2}})$	$\frac{1}{2}e^{\frac{-r}{2}}$
2PSK (BPSK)	$erfc(\sqrt{r})$	$\frac{1}{2}e^{-r}$

**Table 3 biomimetics-09-00103-t003:** Parameter settings for direct sequence spread spectrum signal modulation.

Parameter	Value
Original signal rate	*R*_b_ = 1200 bps
Fifth Order primitive polynomial	polynomial = [1 0 0 1 0 1]
Pseudocode sequence length	*L*_PN_ = 31
Pseudocode rate	*R*_c_ = *L*_PN_ × *R*_b_ = 37,200 chip/s
Sampling frequency	*f*_s_ = 8*R*_c_ = 297,600 Hz
Carrier frequency	*f*_c_ = 50 KHz
Shaping filter roll-off factor	*α* = 0.8

## Data Availability

Data are contained within the article.

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
