# Peer review of "Implementation of Underwater Electric Field Communication Based on Direct Sequence Spread Spectrum (DSSS) and Binary Phase Shift Keying (BPSK) Modulation"

_biomimetics, 2024, doi:10.3390/biomimetics9020103_

Round 1
Reviewer 1 Report
Comments and Suggestions for Authors
This is a very interesting proposal, which addresses an important problem and provides detailed insights into the theoretical and experimental aspects of underwater communications. The reviewer recommends the proposal to be submitted without changes, but he also believes that there are a few questions to which the authors could provide some compact answers, namely: (1) What values of conductivity can be expected for murky waters? (2) Overall, what can be expected to be the value of the signal to noise ratio in the conditions of the experiment? (3) Are signal echoes relevant to the problem of demodulation?
Author Response
Firstly, thank you for your affirmation of our work. Next, I will respond to each of the questions you raised point by point.
Question 1#: What values of conductivity can be expected for murky waters?
Response 1#: The conductivity of turbid water bodies is related to ion concentration and ion movement, and it can be influenced by suspended particulate matter. The conductivity in turbid water bodies can vary widely; when suspended particles hinder the movement of ionized substances, the conductivity of turbid water may be lower. If the water contains dissolved salts or other conductive substances, the conductivity may be relatively high. The general range of conductivity in natural water bodies is between 50 and 1000 uS/cm. In our laboratory pool experiments, we measured a conductivity of 500 uS/cm, while in the swimming pool area, the conductivity was measured at 739 uS/cm.
Question 2#: Overall, what can be expected to be the value of the signal to noise ratio in the conditions of the experiment?
Response 2#:
Under simulation conditions with Gaussian white noise as the noise source, the receiver can achieve error-free demodulation at 0.5 dB. Under experimental conditions, using a logic analyzer to capture digital signals, and considering the impact of thermal noise, electromagnetic radiation, and computing platform capabilities, the receiver's SNR needs to be 3 dB to achieve effective demodulation.
Question 3#: Are signal echoes relevant to the problem of demodulation?
Response 3#: During the process of electric field communication, some signals propagate through a direct path, while others undergo reflection and refraction, forming echoes that superimpose on the receiver. Since the propagation speed of the electric field underwater can be considered similar to the speed of light, the multipath effects generated by the electric field at close distances are not as severe as in underwater acoustic communication. However, the performance of the receiver is still subject to some impact.
Finally, once again, thank you for taking the time to review our manuscript and raise questions.
Reviewer 2 Report
Comments and Suggestions for Authors
The presented scientific article has scientific significance. Providing underwater wireless communications is of great scientific and applied importance.
The results obtained (communication range, data transfer rate) do not allow the proposed system to be used for solving applied problems, but improving the characteristics of the communication system is the subject of further research by the authors.
The scientific article is quite structured. The research methods used are sufficiently described and justified.
Also of interest is the system's noise immunity under different conditions of the aquatic environment, as well as when the signal-to-noise ratio at the receiver input changes. In further research, I suggest that authors also pay attention to these issues.
Author Response
Dear Reviewer:
Thank you for your recognition of our team's efforts and the improvement suggestions. As an experimental paper, we are currently dedicated to enhancing the performance of this communication method for underwater environments. We hope that this biomimetic approach can truly address practical underwater communication challenges.
In the paper, we conducted experiments by varying the relative distance between the receiver and transmitter to validate the system's noise resistance. This allowed us to assess how the system performs when the signal-to-noise ratio at the receiver input undergoes changes.
Conducting experiments in outdoor aquatic conditions is our next planned step.
Once again, we appreciate your time in reviewing our manuscript and posing questions.
All authors
Reviewer 3 Report
Comments and Suggestions for Authors
It is a really interesting paper devoted to a novel promising field. I'm not sure about relevance to scope of Biomimetics, but prefer to leave it to the editor's judgment. For my part, I do not see any significant obstacles to publication, except for poor English (especially usage of terms) and too detailed description of basic physical concepts. It is an experimental work with clear and convincing results. I hope that authors will improve English and add necessary corrections into the descriptive part of the manuscript.
Comments on the Quality of English LanguageThe use of special vocabulary should be corrected.
Author Response
Dear Reviewer:
Thank you for your valuable feedback.,it is very helpful for our paper. We have made the following revisions based on your suggestions. I will now provide a detailed list:
Question 1#: too detailed description of basic physical concepts.
Response 1#:We have appropriately removed redundant physical concepts that are not highly relevant to the paper. Additionally, we have concentrated the content of the first two chapters and significantly revised the content of the third chapter to emphasize the transformation and design of the model.
Question 2#: an experimental work;add necessary corrections into the descriptive part of the manuscript
Response 2#:As you mentioned, since this is an experimental work, we have modified the design of the system in Chapter 3 and supplemented the details of the experiments in Chapter 4.
Question 3#:poor English (especially usage of terms)
Response 3#:We also agree that this is a very serious issue.We have unified the usage of professional terminology,and have made every effort to revise the English expression throughout the entire paper.
The revised version of the paper has been uploaded in PDF format. Once again, thank you for taking the time to review our paper.
All authors

Reviewer 4 Report
Comments and Suggestions for Authors
The paper describes an underwater communication protocol based on electric current field. In my opinion the article is well written, and the research contain interesting results, that can be used in further research and even in applications. I have a few questions and suggestions:
a) In the introduction the authors distinguish between foreign and domestic research results. Since the article is presented in an international journal, I do not understand the reason of this distinction, in my opinion authors should combine the two paragraphs into a single one, with united timeline.
b) The resolution of many figures is too low, thus the text on the figures cannot be distinguished. Such figures are: 3, 4, 10 and 11.
c) The authors tested their method using real experiments. The block diagram of the experiment is given, however there are little details on the instruments and other devices used in the experiment.
d) The authors claim that the power consumption was decreased by 28%. However, there is no data on the power consumption in the article. Personally, I am really interested in this parameter, since the current given in table 1 is quite high.
e) In a real-life communication, how would the current between the dipoles influence the animals at the location. Can this communication be used in saltwater?
Author Response
Dear Reviewer:
Thank you for your valuable feedback,it is very helpful for our paper. We have made the following revisions based on your suggestions. I will now provide a detailed list:
Question 1#: In the introduction the authors distinguish between foreign and domestic research results. Since the article is presented in an international journal, I do not understand the reason of this distinction, in my opinion authors should combine the two paragraphs into a single one, with united timeline.
Response 1#:This issue is indeed crucial, and we appreciate your identification of it. We have made revisions promptly upon reviewing the paper. The decision to publish in an international journal was a spontaneous one, leading to some ambiguities in the paper. We have rewritten and streamlined Chapter 1 accordingly.
Question 2#: The resolution of many figures is too low, thus the text on the figures cannot be distinguished. Such figures are: 3, 4, 10 and 11.
Response 2#:
Without modifying the original content, we made the following adjustments:
- In Figure 3: The theoretical content in the relevant part has been simplified, so Figure 3 has been removed.
- In Figure 4: We have enhanced the resolution as much as possible and enlarged the image.
- Figures 10 and 11: We have created new, more concise, and clearer schematic diagrams to replace them.
Question 3#: The authors tested their method using real experiments. The block diagram of the experiment is given, however there are little details on the instruments and other devices used in the experiment.
Response 3#:
Thank you for your feedback. Regarding your suggestions, we have made specific adjustments in Chapters 3 and 4:
-
In Chapter 3, we present the platforms used, such as FPGA, analog circuits, Raspberry Pi, etc., and provide detailed demonstrations of key technologies, such as the software deployment of pseudocode synchronization on the FPGA.
- In Chapter 4, we have thoroughly reproduced the experimental setup, including the arrangement of electrodes, determination of conductivity, etc. Additionally, we have visualized the digital signal processing of the receiver.
Question 4#:The authors claim that the power consumption was decreased by 28%. However, there is no data on the power consumption in the article. Personally, I am really interested in this parameter, since the current given in table 1 is quite high.
Response 4#:
The power consumption data in the experiment are derived from the power output of the power supply, and we measured this using a power meter instrument during the experiment. The observed reduction in power consumption is in comparison to the team's previous work (refer to Reference 28). In the previous work, the measured power consumption of the transmitter and receiver combined was 8.2 watts, while in this article, the measured overall power consumption of the communication system is 5 watts.
The current parameters provided in Table 1 are aimed at obtaining a more reasonable carrier frequency for simulating electric field communication and are unrelated to the subsequent experiment's power consumption.
Question 5#:In a real-life communication, how would the current between the dipoles influence the animals at the location. Can this communication be used in saltwater?
Response 5#:
The transmission power in the electric field communication system used in this article is relatively low, posing minimal biological threats. Currently, the experiments are conducted in a swimming pool, but it cannot be ruled out that under outdoor conditions, it may have some impact on organisms with charge-sensitive organs.
The system can be used in saltwater, and the conductivity of the swimming pool environment in our recent experiments has already entered the range of saltwater
The revised version of the paper has been uploaded in PDF format. Once again, thank you for taking the time to review our paper.
All authors

Reviewer 5 Report
Comments and Suggestions for Authors
The manuscript describes original research in signal propagation underwater. However to improve the contribution section 3.3 should be enlarged to provide details of the experimental setup. There is no hardware specifications or references. Section 3.3 should be enlarged at least 3 pages, with references. Abstract must describe the main contributions of the manuscript that is in section 3.3. A major revision is needed.
Comments on the Quality of English LanguageEnglish has to be revised ton improve the readibility.
Author Response
Dear Reviewer:
Thank you for your valuable feedback.,it is very helpful for our paper. We have made the following revisions based on your suggestions. I will now provide a detailed list:
Question 1#: section 3.3 should be enlarged to provide details of the experimental setup
Response 1#:Thank you for your feedback. Regarding your suggestions, we have made specific adjustments
- In Chapter 3, we present the platforms used, such as FPGA, analog circuits, Raspberry Pi, etc., and provide detailed demonstrations of key technologies, such as the software deployment of pseudocode synchronization on the FPGA.
- In Chapter 4, we have thoroughly reproduced the experimental setup, including the arrangement of electrodes, determination of conductivity, etc. Additionally, we have visualized the digital signal processing of the receiver.
Question 2#: English has to be revised ton improve the readibility
Response 2#: We have unified the usage of professional terminology,and have made every effort to revise the English expression throughout the entire paper.
Question 3#: References、Methods 、Abstract
Response 3#:We have made modifications to all other sections of the entire paper.
The revised version of the paper has been uploaded in PDF format. Once again, thank you for taking the time to review our paper.
All authors

Round 2
Reviewer 3 Report
Comments and Suggestions for Authors
The paper can be accepted for publication.
Reviewer 5 Report
Comments and Suggestions for Authors
The authors have properly responded to my comments and the manuscript quality has largely improved. I recommend its publication.
Please, make minor corrections to formatting and English.
Comments on the Quality of English LanguagePlease, make minor corrections to formatting and English.